# Effects of 25-Hydroxyvitamin D_3_ and Oral Calcium Bolus on Lactation Performance, Ca Homeostasis, and Health of Multiparous Dairy Cows

**DOI:** 10.3390/ani11061576

**Published:** 2021-05-28

**Authors:** Hongjian Xu, Quanyu Zhang, Lihua Wang, Chengrui Zhang, Yang Li, Yonggen Zhang

**Affiliations:** College of Animal Science and Technology, Northeast Agricultural University, Harbin 150030, China; xuhongjian0714@163.com (H.X.); yugezuishuai1314@yeah.net (Q.Z.); wanglihua4822@163.com (L.W.); Cruivan8254936@163.com (C.Z.); liyang1405053@sina.com (Y.L.)

**Keywords:** subclinical hypocalcemia, milk yield, energy metabolism, oxidative stress

## Abstract

**Simple Summary:**

Subclinical hypocalcemia severely affects the lactation and health of dairy cows. Subclinical hypocalcemia is still a concern with cows after postpartum oral Ca; thus, the single treatment approach gradually has shifted to a multitreatment approach in terms of subclinical hypocalcemia. Supplementing 25-hydroxyvitamin D_3_ could solve the problem of insufficient vitamin D_3_ synthesis and blocked conversion in transition cows. The present study showed that feeding 25-hydroxyvitamin D_3_ combined with oral calcium not only improved serum 25-hydroxyvitamin D_3_ status and calcium homeostasis, but also had potential benefits on lactation performance and the health status during the transition period.

**Abstract:**

Little information is available regarding the effect of supplementing 25-hydroxyvitamin D_3_ during the transition period combined with a postpartum oral calcium bolus on Ca homeostasis. The objectives of the current study were to evaluate the effects of 25-hydroxyvitamin D_3_ combined with postpartum oral calcium bolus on lactation performance, serum minerals and vitamin D_3_ metabolites, blood biochemistry, and antioxidant and immune function in multiparous dairy cows. To evaluate the effects of 25-hydroxyvitamin D_3_ combined with oral calcium, 48 multiparous Holstein cows were randomly assigned to one of four treatments: (1) supplementing 240 mg/day vitamin D_3_ without a postpartum oral Ca bolus (control), (2) supplementing 240 mg/day vitamin D_3_ with an oral Ca bolus containing 90 g of Ca immediately post-calving (Ca + VitD), (3) supplementing 6 g/day 25-hydroxyvitamin D_3_ without an oral Ca bolus (25D), and (4) supplementing 6 g/day 25-hydroxyvitamin D_3_ with an oral Ca bolus containing 90 g of Ca immediately post-calving (Ca + 25D). Lactation performance during the first 21 days was measured. Blood was collected at the initiation of calving and then 1, 2, 7, 14, and 21 days relative to the calving date. The yield of milk (0.05 < *p* < 0.10), energy-corrected milk (*p* < 0.05), 3.5% fat-corrected milk (*p* < 0.05), and milk protein (*p* < 0.05) were significantly higher in 25-hydroxyvitamin D_3_-treated groups within 3 weeks of lactation than in vitamin D_3_-treated cows. The iCa (*p* < 0.05) and tCa (*p* < 0.05) were higher in both Ca and 25D + Ca cows than in the control and 25D groups within 48 h. The concentrations of serum tCa (*p* < 0.05), tP (*p* < 0.05), and 25-hydroxyvitamin D_3_ (*p* < 0.05) in 25D and 25D + Ca cows were higher than those in control and Ca cows within 21 days postpartum. Feeding 25-hydroxyvitamin D_3_ also showed a lower concentration of malondialdehyde (*p* < 0.05), interleukin 6 (*p* < 0.05), and tumor necrosis factor-alpha (TNF-α) (*p* < 0.05), as well as a higher concentration of alkaline phosphatase (*p* < 0.05), total antioxidant capacity (*p* < 0.05), and immunoglobulin G (*p* < 0.05) than vitamin D_3_. Supplementing Ca bolus also showed lower concentrations of alanine transaminase (*p* < 0.05) and TNF-α (*p* < 0.05). In conclusion, supplementing 25-hydroxyvitamin D_3_ during the transition period combined with a postpartum oral calcium bolus improved lactation performance, Ca homeostasis, and antioxidant and immune function of medium-production dairy cows within 21 days postpartum.

## 1. Introduction

During the transition period, hypocalcemia severely affects the health, lactation performance, and reproductive performance of dairy cows [1]. In addition to decreased peripartal dry matter intake (DMI), hypocalcemia directly leads to milk fever and triggers other health disorders in cows, such as retained placenta, ketosis, displaced abomasum, metritis, and mastitis. Hypocalcemia was caused mainly by an abruptly increased Ca demand of colostrogenesis and subsequent milk production and delayed or disordered Ca mobilization. Strategies that could improve calcium mobilization or increase the blood concentration of calcium should be able to prevent hypocalcemia.

In recent years, increased attention has been directed toward postpartum oral Ca boluses for preventing the occurrence of hypocalcemia. This strategy of Ca delivery is feasible because Ca is absorbed by passive diffusion through the junctions connected with mucosal cells that depend on Ca concentrations in the lumen [2]. Thus, increasing the Ca concentration in the rumen using Ca supplementation is important to enhance Ca absorption. However, supplementing oral Ca effectively increases blood Ca concentration only within 48 h after calving [3,4]. Cows remained at the risk of subclinical hypocalcemia during the postpartum period. Thus, enhancing internal Ca mobilization is still necessary for transition cows.

Vitamin D_3_ participates in Ca homeostasis by manipulating Ca mobilization of bone, reabsorption of the kidney, and absorption from the intestine [1]. Because inadequate 1,25-dihydroxyvitamin D_3_ synthesis or sensitivity after calving in part contributes to hypocalcemia, the use of vitamin D_3_ metabolites with biological activity provides an appealing opportunity to stimulate mechanisms of Ca mobilization. Indeed, transition cows have a limited capacity to convert vitamin D_3_ to 25-hydroxyvitamin D_3_. The application of 25-hydroxyvitamin D_3_ has effectively increased serum 25-hydroxyvitamin D_3_ in transition dairy cows [5]. Moreover, feeding 25-hydroxyvitamin D_3_ during late gestation has increased the prepartum concentrations of 1,25-dihydroxyvitamin D_3_, ionized Ca, total Ca, and P in plasma, but these increases were not observed during the postpartum period [6]. Thus, supplementing 25-hydroxyvitamin D_3_ during the whole transition period has the potential to increase the concentration of serum 1,25-dihydroxyvitamin D_3_ and tCa in postpartum cows. In addition, feeding 25-hydroxyvitamin D_3_ was shown to improve the yield and composition of colostrum compared with feeding vitamin D_3_, but resulted in a greater loss of Ca and Mg in colostrum and a negative energy balance [7]. Oral Ca supplementation has the potential to offset the extra loss of Ca caused by feeding 25-hydroxyvitamin D_3_ in time. Previous studies reported that 25-hydroxyvitamin D_3_ possibly benefited the antioxidant and immune systems [8,9]. Feeding 25-hydroxyvitamin D_3_ also improved the oxidative burst activity of neutrophils and decreased the incidences of multiple diseases in dairy cows during early lactation [6]. Nevertheless, the effects of postpartum Ca supplementation combined with 25-hydroxyvitamin D_3_ during the whole transition period on Ca homeostasis, as well as the antioxidant and immune functions in cows, are not well defined.

Therefore, this study aims to evaluate the effects of feeding 25-hydroxyvitamin D_3_ during the transition period in combination with postpartum oral Ca on lactation performance, minerals and vitamin D_3_ metabolites, blood biochemistry, and antioxidant and immune function. We hypothesized that feeding 25-hydroxyvitamin D_3_ during the transition period in combination with postpartum oral Ca would increase serum Ca within 48 h and 21 days postpartum, and would have the potential to improve lactation performance and antioxidant and immune functions compared with postpartum oral Ca supplements alone and the control.

## 2. Materials and Methods

All experimental procedures involving animals were performed according to the principles outlined in the Northeast Agricultural University Animal Care and Use Committee’s guidelines (protocol number: NEAU-[2011]-9).

### 2.1. Animals, Diets, and Experimental Design

This study was conducted at the Songhuajiang Dairy Farm (Harbin, Heilongjiang Province, China). Overall, 48 dry Holstein cows (2–4 parity) were selected on the basis of lactation numbers, anticipated calving dates, and milk yields in the previous lactation. Cows were moved from the dry cow pen into individual pens 21 d before anticipated calving and fed prepartum diets. After one week of adaptation, 48 Holstein cows were randomly assigned to 4 groups based on body condition, gestation numbers, anticipated calving dates, and milk yields in previous lactation. Treatments were (1) supplementing 240 mg/day vitamin D_3_ without a postpartum oral Ca bolus (control), (2) supplementing 240 mg/day vitamin D_3_ with an oral Ca bolus containing 90 g of Ca immediately post-calving (Ca + VitD), (3) supplementing 6 g/day 25-hydroxyvitamin D_3_ without an oral Ca bolus (25D), and (4) supplementing 6 g/day 25-hydroxyvitamin D_3_ with an oral Ca bolus containing 90 g of Ca immediately post-calving (Ca + 25D). To make cows consume all vitamin D_3_ or 25-hydroxyvitamin D_3_ concentrates readily after delivery, a top-dress supplement was prepared for each cow by mixing vitamin D_3_ or 25-hydroxyvitamin D_3_ concentrates with finely ground corn in 100 g of the mixture. Rations were subsequently fed to individual cows. Cows were fed a total mixed ration (TMR) once daily at 07:30 before calving and twice a day at 07:00 and 15:00 in the postpartum period. The vitamin D_3_ product contained 500 KIU/g vitamin D_3_ (Zhongmu Biological Pharmaceutical Co., Ltd., Zhengzhou, China), and the 25D product contained 20 KIU/g 25-hydroxyvitamin D_3_ (Haineng Biological Engineering Co. Ltd., Rizhao, China). An oral Ca bolus containing 90 g of Ca was provided by Qilu Animal Health Products Co. Ltd. (Jinan, Shandong, China). Diets were formulated to meet or exceed the nutrient requirements of cows (National Research Council) for both the late gestation and lactation periods (Table 1) [10].

### 2.2. Sample Collection, Measurements, and Analyses

DMI was recorded daily by recording the amount of feed offered and left and adjusted daily to result in at least 5% refusals. Feed samples and orts were collected weekly for wet chemistry analysis. The feed and orts samples were dried at 55 °C for 48 h in an air-forced oven, and moisture loss was recorded. Dried samples were ground to pass a 1 mm screen of a Wiley mill, and analyzed for DM (105 °C for 12 h). The dry matter (DM) (method 930.15), the CP (method 984.13), ether extract (method 920.39), and minerals (method 985.01) were determined using the techniques specified by the Association of Official Analytical Chemists [11]. NDF was quantified according to the method [12]. Starch content was measured using the Megazyme Total Starch Assay Kit (product no: K-TSTA; Megazyme International Ireland Ltd., Wicklow, Ireland). Body condition was scored on the day of enrollment and then once weekly until 21 DIM by the same two trained evaluators using a 5-scale system and the average was used for statistical analysis [13].

### 2.3. Measurements of Milk and Milk Components

All cows were housed in individual pens until 21 days in milk (DIM). Cows were milked two times a day at 06:00 and 17:00, and milk yield (the sum of two milkings) was recorded until 21 d post-calving. Milk samples were sampled on 2 consecutive days on days 6, 7, 13, 14, 20, and 21. A 24-h composite sample was mixed according to the actual volume of milk yield of day and night. Milk samples were analyzed for fat, true protein, lactose, and SCC. Yields of milk corrected for 3.5% fat content and for energy were calculated according to the NRC as follows:3.5% FCM = 0.4324 × milk kg + (16.218 × milk fat kg)(1)
ECM = [(0.3246 × milk yield) + (12.86 × fat yield) + (7.04 × protein yield)](2)
SCS = log_2_ (somatic cell counts/100) + 3(3)

### 2.4. Blood Collection and Analysis

Whole blood, serum and plasma were collected into evacuated tubes at 0, 1, 2, 7, 14, and 21 d relative to calving by puncture of the coccygeal blood vessels. Evacuated tubes containing no anticoagulant agents were separated for serum and those containing K_2_EDTA for plasma.

Whole blood was analyzed within 1 to 3 min for concentrations of iCa using a handheld biochemical analyzer (Harold Technology Co. Ltd., Beijing, China). The concentrations of total Ca (tCa), total P (tP), total Mg (tMg), and 25-hydroxyvitamin D_3_ were analyzed in serum samples collected on days 0, 1, 2, 7, 14, and 21 relative to calving in duplicate. Samples were analyzed for tCa and tMg by atomic absorption. Concentrations of tP were quantified in serum using the molybdenum blue method. Serum samples were analyzed for the concentrations of 25-hydroxyvitamin D_3_ using validated ELISA methods (HY-NE100) (SINO-UK Institute of Biotechnology, Beijing, China). The plasma concentrations of total protein (TP), glucose, triglyceride (TG), cholesterol, β-hydroxybutyric acid (BHB), non-esterified fatty acid (NEFA), alanine transaminase (ALT), aspartate aminotransferase (AST), and alkaline phosphatase (ALP) were measured using a fully automatic biochemical analyzer provided by Biosino Biotech (Beijing).

Total superoxide dismutase (T-SOD, method: hydroxylamine; kit number: A001-1-2), total antioxidant capacity (T-AOC, method: 2, 2 0 -azino-bis (3-ethylbenzothiazoline-6-sulfonic acid); kit number: A015-1-2), catalase (CAT, method: visible light; kit number: A007-1-1), and malondialdehyde (MDA, method: thiobarbituric; kit number: A003-1-2) in plasma were determined by colorimetric analysis kits (Nanjing Jiancheng Institute of Bioengineering, Nanjing, China). The determination of IgA, IgG, and IgM levels was conducted by ELISA using bovine IgA/G/M ELISA kits (HY-50093) (SINO-UK Institute of Biotechnology, Beijing, China). The concentrations of cytokines (interleukin-6 (IL-6, method: liquid-phase competition; kit number: HY-10116) and tumor necrosis factor-α (TNF-α, method: liquid-phase competition; kit number: HY-10105)) were measured with bovine RIA kits (SINO-UK Institute of Biotechnology, Beijing, China).

### 2.5. Statistical Analysis

The data were analyzed by SAS 9.4 software (SAS Institute Inc., Cary, NC, USA) using the MIXED procedure (PROC MIXED), which included repeated measures (DMI, BCS, milk yield and compositions, feed efficiency, SCS, and blood metabolites). The source of vitamin D_3_ (vitamin D_3_ vs. 25-hydroxyvitamin D_3_), Ca bolus (without oral Ca bolus vs. with oral Ca bolus), source × Ca, time, source × time, Ca × time, and source × Ca × time were considered as fixed effects, and the cow was considered as a random effect. Initial variables were used as covariates for analyses. Data were reported as least square means and were considered significant if *p* ≤ 0.05 and as a tendency if 0.05 < *p* ≤ 0.10. When the interaction between source × Ca was significant, the SLICE option in the LSMEANS statement was used to determine differences among the factors.

## 3. Results

### 3.1. Lactation Performance

Dry matter intake was not influenced (*p* > 0.10; Table 2) by 25-hydroxyvitamin D_3_ and Ca bolus treatments during the 3 weeks after calving. Cows treated with 25-hydroxyvitamin D_3_ produced more milk yield (0.05 < *p* < 0.10), 3.5% FCM (*p* ≤ 0.05), ECM (*p* < 0.05), protein content (*p* < 0.05), protein yield (*p* < 0.05), and fat yield (0.05 < *p* < 0.10) in comparison with cows fed vitamin D_3_ during the 21 days after calving. The SCS in milk, BCS, and feed efficiency were not affected by treatment (*p* > 0.10). 

### 3.2. Serum Minerals and Vitamin D_3_ Metabolites

Postpartum oral Ca bolus increased blood iCa (*p* < 0.05) and serum concentration of tCa (*p* < 0.05) within 48 h regardless of the source of vitamin D_3_ (Table 3). Feeding dietary 25-hydroxyvitamin D_3_ increased serum 25-hydroxyvitamin D_3_ (*p* < 0.05) compared with vitamin D_3_ within 48 h. Feeding dietary 25-hydroxyvitamin D_3_ significantly increased blood iCa (*p* < 0.05) and serum concentrations of tCa (*p* < 0.05), tP (*p* < 0.05), and 25-hydroxyvitamin D_3_ (*p* < 0.05) compared with vitamin D_3_ throughout 21 DIM.

### 3.3. Blood Biochemistry

Table 4 shows the concentrations of energy and liver metabolites during the first 21 days post-calving. The treatments did not affect concentrations of glucose, total protein, TG, cholesterol, NEFAs, BHB, or AST in plasma (*p* > 0.10). Cows supplemented with an oral Ca bolus had lower ALT than cows without supplementing Ca bolus (*p* < 0.05). In addition, a significantly higher concentration of ALP (*p* < 0.05) was observed in the cows fed 25-hydroxyvitamin D_3_ than vitamin D_3_.

### 3.4. Antioxidant and Immune Functions

Lower production of MDA (*p* < 0.05) was observed in cows fed 25-hydroxyvitamin D_3_ compared to vitamin D_3_ groups (Table 5).The T-AOC (*p* < 0.05) and IgG level (*p* < 0.05) were higher in the 25D groups than in the vitamin D_3_ groups. Cows receiving 25-hydroxyvitamin D_3_ showed significantly lower IL-6 (*p* < 0.05) and TNF-α (*p* < 0.05) than cows receiving the vitamin D_3_ diet. Cows supplemented with calcium bolus had a lower concentration of TNF-α (*p* < 0.05) compared with cows without calcium bolus supplementation.

## 4. Discussion

Subclinical hypocalcemia reduces DMI and leads to negative energy balance after calving, thus impairing lactation performance and causing postpartum health disorders [1]. Although an acidogenic diet, low Ca diets and postpartum oral bolus Ca supplements have decreased the incidences of clinical hypocalcemia to a large extent, subclinical hypocalcemia has remained a prevalent issue in the cattle industry [2].

In the present study, supplementing cows with an oral Ca bolus or 25-hydroxyvitamin D_3_ did not affect DMI and BCS during the postpartum period in the present study. Sufficient concentrations of blood iCa and serum tCa probably reduced the incidence of subclinical hypocalcemia. Subclinical hypocalcemia changed with the threshold selected, and subclinical hypocalcemia might have happened in control cows when the threshold was iCa ≤ 1.0 mM and tCa ≤ 2.0 mM. Nevertheless, when the threshold was tCa < 2.15 mM, both the control and 25D cows had the potential to be diagnosed with subclinical hypocalcemia within 24 h postpartum. Subclinical hypocalcemia was the important reason of depression of gastrointestinal movement [1]. Cows with oral Ca supplementation were reported to produce more milk driven by more DMI [14].

Supplementing cows with 25-hydroxyvitamin D_3_ increased milk yield because feeding transition cows with 25-hydroxyvitamin D_3_ further increased the blood iCa and tCa concentrations compared with the other groups during 21 DIM postpartum, thereby further reducing the incidence of subclinical hypocalcemia. Decreased incidences of subclinical hypocalcemia might positively affect lactation performance because subclinical hypocalcemia associated with other postpartum diseases caused substantial limitations to milk performance. A previous experiment also observed increased colostrum and milk yield along with reduced morbidity by feeding dietary 25-hydroxyvitamin D_3_ in dairy cows [7]. Next, feeding 25-hydroxyvitamin D_3_ might increase the expression of 1,25-dihydroxyvitamin D_3_ in mammary epithelial cells in transition cows. It was reported that 1,25-dihydroxyvitamin D_3_ benefited the proliferation and differentiation of mammary epithelial cells, thereby improving lactation performance [15]. Poindexter et al. also reported the effects of feeding 25-hydroxyvitamin D_3_ on mammary immunity and supported a role for vitamin D signaling in protection of the mammary gland from mastitis [16]. This effect can also explain the increase in milk protein production in the present study. In addition, the positive effects of feeding 25-hydroxyvitamin D_3_ on antioxidant and immune function in the present study also contributed to better milk production and milk protein. During the oxidative stress and inflammatory phases, there was a shift in nutrient partitioning to responses involved in survival instead of lactation.

A postpartum oral Ca bolus can effectively increase the postpartum blood calcium concentration within 48 h. The increase in blood calcium concentration was dependent on rumen Ca transportation when the bolus supplement contained 40 g of Ca in a rumen volume of 100 L [17]. Supplementing 25-hydroxyvitamin D_3_ did not further increase the blood calcium concentration within 48 h, which may be due to the temporary inhibition of the function of 1,25-dihydroxyvitamin D_3_ in regulating blood calcium after calving. In contrast, feeding 25-hydroxyvitamin D_3_ combined with a low DCAD diet maintained high iCa and tCa concentrations in the 24 h following parturition [18]. Supplementing 25-hydroxyvitamin D_3_ in place of vitamin D_3_ during the transition period was superior in increasing postpartum plasma concentrations of 25-hydroxyvitamin D_3_, 1,25-dihydroxyvitamin D_3_, and the concentrations of tCa and tP in dairy cows at 3 weeks postpartum. Although clinical hypocalcemia occurs within 48 h after calving, subclinical hypocalcemia threatens the performance and health of dairy herds throughout the postpartum period [19]. To date, the effects of feeding 25-hydroxyvitamin D_3_ during the whole transition period on Ca homeostasis postpartum have not been well defined. Feeding 25-hydroxyvitamin D_3_ postpartum increased calcium in plasma and milk compared to prepartum feeding [5]. When ceasing 25-hydroxyvitamin D_3_ supplementation at calving, previous studies showed a persistent decline in postpartum serum 25-hydroxyvitamin D_3_ [6,20]. Thus, postpartum feeding 25-hydroxyvitamin D_3_ was important to maintain serum concentrations of 25-hydroxyvitamin D_3_ and tCa during the postpartum period due to higher efficiency of the conversion to 1,25-dihydroxyvitamin D_3_. Adequate serum concentrations of 25-hydroxyvitamin D_3_ may contribute to better lactation performance and health status. Our results suggested that 25-hydroxyvitamin D_3_ can be used in conjunction with an oral Ca bolus to potentially improve tCa during the first 3 weeks postpartum compared with a Ca bolus alone. The oral Ca bolus did not affect serum P, but when combined with 25-hydroxyvitamin D_3_, serum P levels increased in cows. The treatments had no adverse effects on serum Mg levels in the present study.

During the transition period, hypocalcemia depressed DMI and led to a negative energy balance. Subclinical hypocalcemia also induced increased lipid mobilization. Excessive lipid mobilization led to increased production of NEFAs and BHB. Elevated serum concentrations of BHB may trigger hyperketonemia. In addition, Ca supplementation tended to increase the concentrations of fatty acids in serum [3]. Feeding 25-hydroxyvitamin D_3_ tended to increase the concentrations of cholesterol and BHB [7]. In the present study, the treatments did not influence plasma concentrations of energy metabolites, implying that higher milk production caused by treatments did not lead to more serious negative energy balance. The current results found that a postpartum oral Ca bolus combined with vitamin D_3_ or 25-hydroxyvitamin D_3_ decreased the level of ALT. The levels of AST and ALT in blood are greater when the liver suffers pathological or chemical injury. In our study, decreased ALT levels suggested that prevention of subclinical hypocalcemia could have a protective effect on hepatic cells. In addition, we found that the ALP level was significantly higher in cows fed 25-hydroxyvitamin D_3_ compared to vitamin D_3_. Most of the serum ALP originates from bone tissues, and higher serum ALP levels indicate greater calcium deposits in bone during growth spurts in humans and the suckling stage of calves [21]. The reason for the increase in ALP in this study may be due to 25-hydroxyvitamin D_3_ increasing bone calcium deposition. Rodney et al. (2017) also reported that feeding 25-hydroxyvitamin D_3_ increased bone accretion relative to resorption by changing bone markers, which might have benefited the bone health of dairy cows [6].

Dairy cows experience tremendous physiological stress around calving. Excessive generation of reactive oxygen species (ROS) induces cellular lipid peroxidation and damage to tissue and cellular functions [8]. Feeding 25-hydroxyvitamin D_3_ increased the T-AOC, which eliminated the overproduction of ROS and reduced the risk of accumulation of oxidative damage in the cows. Supplementation of 25-hydroxyvitamin D_3_ enhanced GSH-Px activity in weaned piglets [22]. The present study showed that feeding 25-hydroxyvitamin D_3_ may reduce the concentration of MDA, indicated that there was a positive effect of 25-hydroxyvitamin D_3_ on lipid peroxidation. The 1,25-dihydroxyvitamin D_3_ could be a potential membrane antioxidant as it protects the membrane from lipid peroxidation through its hydrophobic parts [23,24]. The compromised antioxidant capacity also leads to immune and inflammatory dysfunction [25]. Feeding 25-hydroxyvitamin D_3_ increased IgG during the postpartum period. The possible reason was that 1,25-dihydroxyvitamin D_3_ stimulated B cells to produce more IgG. As the ultimate active vitamin D_3_, 1,25-dihydroxyvitamin D_3_ also modulates innate and adaptive immunity in cattle, and B cells are a significant source of 1,25-dihydroxyvitamin D_3_ in the bovine immune system [9]. The 1,25-dihydroxyvitamin D_3_ was reported to upregulate IL-10, which further promoted plasmablasts to secrete immunoglobulin [26,27]. Another possible reason is that calcium is also recognized as an important signaling molecule in B cell function that could influence IgG production [28]. Feeding 25-hydroxyvitamin D_3_ also increased the concentrations of IgG in colostrum [7]. Pro-inflammatory cytokine assessment provides information on adaptive immune responses in cattle. In the present study, feeding 25-hydroxyvitamin D_3_ decreased the concentrations of IL-6 and TNF-α during the postpartum period, which was perhaps because of the depression of systemic inflammatory reactions. Previous studies evaluating the effects of 1,25-dihydroxyvitamin D_3_ on adaptive immune responses in cattle indicated that 1,25-dihydroxyvitamin D_3_ inhibited pro-inflammatory interferon (IFN)-γ and interleukin (IL)-17 responses [29,30]. Among the biological functions of IFN-γ is to stimulate the secretion of IL-1, IL-6, IL-8, and TNF-a. Therefore, the decrease in TNF-α may be due to the effect of 25-hydroxyvitamin D_3_ on adaptive immune responses in cattle. Sufficient concentrations of serum 1,25-dihydroxyvitamin D_3_ become critical and necessary for an optimal response. Pro-inflammatory responses also occurred when calves were infected with bovine diarrhea virus, which dramatically decreased the serum concentration of 25-hydroxyvitamin D_3_. The virus infection increased the 1α-hydroxylase (vitamin D-activating enzyme) expression of immune cells for catalyzing the conversion of 25-hydroxyvitamin D_3_ to 1,25-dihydroxyvitamin D_3_ to increase the antibacterial capacity and immune responses associated with the dampening of excessive inflammatory reactions [31]. The potentially beneficial immunomodulatory effects of 25-hydroxyvitamin D_3_ in cattle may reduce reliance on antimicrobials, improve the health and productivity of cattle, and as a consequence improve food safety; thus, dietary 25-hydroxyvitamin D_3_ warrants continued investigation.

## 5. Conclusions

Supplementing an oral Ca bolus immediately post-calving during the transition period increased the blood iCa and serum tCa levels during 48 h, and cows fed 25-hydroxyvitamin D_3_ showed higher serum tCa, tP, and 25-hydroxyvitamin D_3_ levels during the first 21 DIM compared with vitamin D_3_. Feeding 25-hydroxyvitamin D_3_ improved the milk yield and milk protein of medium-production dairy cows. Feeding 25-hydroxyvitamin D_3_ potentially improved the antioxidant capacity and IgG concentration, and resulted in a reduction in inflammation. Our results indicated that supplementing with an oral Ca bolus immediately post-calving combined with 25-hydroxyvitamin D_3_ during the transition period improved Ca homeostasis, 25-hydroxyvitamin D_3_ status, and lactation performance, and had the potential to benefit health through the depression of oxidative stress and inflammatory reactions and increment of IgG.

## Figures and Tables

**Table 1 animals-11-01576-t001:** Dietary ingredients and nutrient compositions of diets fed prepartum and postpartum.

Item	Prepartum	Postpartum
Ingredient, % of DM		
Corn silage	57.9	29.0
Alfalfa hay	—	24.5
Wheat straw	10.8	3.5
Chinese wildrye	9.7	—
Steam-flaked corn	4.8	19.8
Soybean meal	11.3	16.3
Whole cottonseed	3.2	3.6
Calcium bicarbonate	1.1	1.5
Sodium bicarbonate	—	0.5
Magnesium oxide	0.5	0.5
Salt	0.3	0.3
Vitamin–mineral mix ^1^	0.3	0.5
Nutrient composition, % of DM		
NE_L_, Mcal/kg	1.57	1.64
DM	49.3	50.4
CP	13.8	18
NDF	44.5	33.6
EE	3.2	3.5
Starch	22.6	25.9
Ca	0.58	0.92
P	0.32	0.35
Mg	0.31	0.35

^1^ Preparation contained (per kg of DM) 3,000,000 IU vitamin A, 25,000 IU vitamin E, 4.25 g Cu, 0.1 g Co, 0.24 g I, 0.20 g Fe, 3.5 g Mn, 0.1 g Se, and 14 g Zn per kg. NE_L_: net energy for lactation; DM: dry matter; CP: crude protein; NDF: neutral detergent fiber; EE: ether extract.

**Table 2 animals-11-01576-t002:** Effect of 25-hydroxyvitamin D_3_ and oral calcium bolus on lactation performance of postpartum Holstein cows.

Items	Treatment ^1^				SEM	*p* Value		
	Control	Ca + VitD	25D	Ca + 25D		VD ^2^	Ca	VD × Ca
DMI ^3^, kg/d	16.5	17.4	17.0	17.8	0.534	0.334	0.119	0.891
BCS	2.87	2.90	2.92	2.94	0.058	0.374	0.678	0.953
Milk Yield, kg/d	21.3	21.6	22.3	23.1	0.711	0.060	0.413	0.683
3.5% FCM, kg/d	25.4	26.2	27.1	28.3	0.946	0.050	0.298	0.833
ECM, kg/d	25.1	26.1	27.3	28.5	0.914	0.014	0.251	0.880
FE(ECM/DMI)	1.59	1.61	1.63	1.66	0.114	0.608	0.499	0.354
Milk composition								
Fat	4.74	4.85	4.87	4.88	0.092	0.429	0.511	0.581
True protein	3.23	3.36	3.53	3.58	0.065	<0.001	0.164	0.608
Lactose	4.56	4.60	4.58	4.57	0.045	0.770	0.785	0.572
Yield, kg/d								
Fat	1.00	1.04	1.08	1.13	0.041	0.056	0.272	0.917
True protein	0.68	0.72	0.79	0.82	0.028	<0.001	0.193	0.956
Lactose	0.97	1.00	1.02	1.06	0.035	0.118	0.337	0.767
SCS	4.17	4.03	3.98	3.89	0.126	0.191	0.375	0.853

^1^ Treatments were (1) supplementing 240 mg/day vitamin D_3_ without a postpartum oral Ca bolus (control), (2) supplementing 240 mg/day vitamin D_3_ with an oral Ca bolus containing 90 g of Ca immediately post-calving (Ca + VitD), (3) supplementing 6 g/day 25-hydroxyvitamin D_3_ without an oral Ca bolus (25D), and (4) supplementing 6 g/day 25-hydroxyvitamin D_3_ with an oral Ca bolus containing 90 g of Ca immediately post-calving (Ca + 25D). ^2^ VD = the effect of the source of vitamin D_3_; Ca = the effect of oral Ca bolus. ^3^ DMI = dry matter intake; BCS = body condition score; FCM = fat-corrected milk; ECM = energy-corrected milk; FE = feed efficiency; SCS = somatic cell scores.

**Table 3 animals-11-01576-t003:** Effect of 25-hydroxyvitamin D_3_ and oral calcium bolus on serum minerals and vitamin D_3_ metabolites of postpartum Holstein cows.

Items	Treatment ^1^				SEM	*p* Value		
	Control	Ca + VitD	25D	25D + Ca		VD ^2^	Ca	VD × Ca
Within 48 h								
Ionized Ca, mM	0.95	1.18	1.09	1.22	0.078	0.351	0.025	0.485
Total Ca, mM	1.90	2.17	2.02	2.20	0.045	0.114	<0.001	0.251
Total P, mM	1.55	1.62	1.64	1.66	0.058	0.240	0.464	0.729
Total Mg, mM	1.01	0.99	1.01	1.00	0.022	0.799	0.513	0.867
25-hydroxyvitamin D_3_, ng/mL	54.5	48.9	130.2	128.4	4.114	<0.001	0.373	0.651
Within 3 weeks								
Ionized Ca, mM	1.06	1.16	1.23	1.25	0.038	<0.001	0.123	0.328
Total Ca, mM	2.22	2.27	2.41	2.48	0.046	<0.001	0.181	0.968
Total P, mM	1.68	1.69	1.81	1.93	0.055	0.001	0.255	0.360
Total Mg, mM	1.03	1.04	1.03	1.05	0.023	0.803	0.627	0.979
25-hydroxyvitamin D_3_, ng/mL	42.8	49.8	150.2	157.1	4.740	<0.001	0.143	0.991

^1^ Treatments were (1) supplementing 240 mg/day vitamin D_3_ without a postpartum oral Ca bolus (control), (2) supplementing 240 mg/day vitamin D_3_ with an oral Ca bolus containing 90 g of Ca immediately post-calving (Ca + VitD), (3) supplementing 6 g/day 25-hydroxyvitamin D_3_ without an oral Ca bolus (25D), and (4) supplementing 6 g/day 25-hydroxyvitamin D_3_ with an oral Ca bolus containing 90 g of Ca immediately post-calving (Ca + 25D). ^2^ VD = the effect of the source of vitamin D_3_; Ca = the effect of oral Ca bolus.

**Table 4 animals-11-01576-t004:** Effect of 25-hydroxyvitamin D_3_ and oral calcium bolus on blood biochemistry of postpartum Holstein cows.

Items	Treatment ^1^				SEM	*p* Value		
	Control	Ca + VitD	25D	Ca + 25D		VD ^2^	Ca	VD × Ca
Glucose, mM	3.53	3.86	3.71	3.84	0.197	0.684	0.257	0.616
TP ^3^, g/L	64.8	66.5	64.5	67.0	1.577	0.967	0.183	0.805
TG, mM	0.24	0.26	0.24	0.26	0.011	0.857	0.128	0.966
Cholesterol, mM	2.19	2.08	2.07	1.90	0.088	0.107	0.118	0.742
NEFA, mM	0.49	0.43	0.41	0.39	0.038	0.138	0.302	0.528
BHB, mM	0.72	0.70	0.70	0.67	0.039	0.613	0.469	0.895
ALT, U/L	24.7	21.4	23.7	20.2	0.899	0.204	<0.001	0.882
AST, U/L	50.9	46.2	49.1	46.9	2.346	0.808	0.143	0.600
ALP, U/L	33.0	34.8	37.1	39.4	1.718	0.014	0.238	0.883

^1^ Treatments were (1) supplementing 240 mg/day vitamin D_3_ without a postpartum oral Ca bolus (control), (2) supplementing 240 mg/day vitamin D_3_ with an oral Ca bolus containing 90 g of Ca immediately post-calving (Ca + VitD), (3) supplementing 6 g/day 25-hydroxyvitamin D_3_ without an oral Ca bolus (25D), and (4) supplementing 6 g/day 25-hydroxyvitamin D_3_ with an oral Ca bolus containing 90 g of Ca immediately post-calving (Ca + 25D). ^2^ VD = the effect of the source of vitamin D_3_; Ca = the effect of oral Ca bolus. ^3^ TP: total protein; TG: triglyceride; NEFA: non-esterified fatty acid; BHB: β-hydroxybutyric acid; ALT: alanine transaminase; AST: aspartate transaminase; ALP: alkaline phosphatase.

**Table 5 animals-11-01576-t005:** Effect of 25-hydroxyvitamin D_3_ and oral calcium bolus on antioxidant and immune function of postpartum Holstein cows.

Items	Treatment ^1^				SEM	*p* Value		
	Control	Ca + VitD	25D	Ca + 25D		VD ^2^	Ca	VD × Ca
Antioxidant								
CAT ^3^, U/mL	55.3	56.8	56.6	58.3	2.194	0.463	0.513	0.965
T-SOD, U/ mL	59.5	60.5	60.2	61.8	1.791	0.593	0.453	0.872
T-AOC, U/mL	11.0	11.6	12.5	12.8	0.406	<0.001	0.282	0.738
MDA, nmol/ mL	3.31	3.13	2.77	2.60	0.192	0.007	0.377	0.951
Immunoglobulins								
Ig A, g/L	0.60	0.61	0.61	0.63	0.040	0.748	0.599	0.918
Ig G, g/L	12.9	13.7	14.6	14.9	0.468	0.003	0.235	0.617
Ig M, g/L	3.13	3.45	3.47	3.68	0.186	0.126	0.155	0.783
Cytokines								
IL-6, pg/ mL	139.9	129.2	121.2	125.0	5.101	0.015	0.658	0.241
TNF-α, pg/ mL	73.0	64.9	63.0	62.1	2.014	0.004	0.051	0.134

^1^ Treatments were (1) supplementing 240 mg/day vitamin D_3_ without a postpartum oral Ca bolus (control), (2) supplementing 240 mg/day vitamin D_3_ with an oral Ca bolus containing 90 g of Ca immediately post-calving (Ca + VitD), (3) supplementing 6 g/day 25-hydroxyvitamin D_3_ without an oral Ca bolus (25D), and (4) supplementing 6 g/day 25-hydroxyvitamin D_3_ with an oral Ca bolus containing 90 g of Ca immediately post-calving (Ca + 25D). ^2^ VD = the effect of the source of vitamin D_3_; Ca = the effect of oral Ca bolus. ^3^ CAT: catalase; T-SOD: total superoxide dismutase; T-AOC: total antioxidant capacity; MDA: malondialdehyde; IgA: immunoglobulin A; IgG: immunoglobulin G; IgM: immunoglobulin M; IL-6: interleukin 6; TNF-α: tumor necrosis factor-α.

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
