# Peer review of "Effects of 25-Hydroxyvitamin D3 and Oral Calcium Bolus on Lactation Performance, Ca Homeostasis, and Health of Multiparous Dairy Cows"

_animals, 2021, doi:10.3390/ani11061576_

Round 1
Reviewer 1 Report
The paper presents interesting results for the interesting topic of vit D and calcium on managing the risk of hypocalcemia in transition dairy cattle. The paper was well written and designed and I have only some minor suggestions for improvement of the manuscript.
L64 - participates
L103&107-7 and elsewhere for consistency: The control is described as "without a postpartum oral calcium bolus" which is clear. I think treatment 3 could be described similarly by saying "without an oral calcium bolus." I think it over-complicates things to say "without an oral calcium bolus containing 90g of calcium...." In my mind it's simpler and less confusing to the reader. This approach could be applied to the relevant Table footnotes as well.
L112 reword: Rations were subsequently fed to individual cows.
L242 typo: without
L262 - delete "and"; perhaps some indication of normal blood calcium concentration in the sentence to provide context with respect to subclinical hypocalcemia threshold.
L296 suggest replacing delivery with calving.
Tables: the treatment headings could be clearer for readers I think. My suggestion would be: Control, Ca+VitD, 25D and Ca+25D
Author Response
Reviewers' comments:
Reviewer: 1
Comment: The paper presents interesting results for the interesting topic of vit D and calcium on managing the risk of hypocalcemia in transition dairy cattle. The paper was well written and designed and I have only some minor suggestions for improvement of the manuscript.
Answer: According to yours comments and suggestions, we have revised the manuscript carefully. In the following pages are our point-by-point responses to the reviewers’ comments/suggestions.
C: L64 – participates
A: Thank you for pointing out the problem. We are very sorry for our carelessness. The “particated” has been revised to “particates” in the revised manuscript.
C: L64 - L103&107-7 and elsewhere for consistency: The control is described as "without a postpartum oral calcium bolus" which is clear. I think treatment 3 could be described similarly by saying "without an oral calcium bolus." I think it over-complicates things to say "without an oral calcium bolus containing 90g of calcium...." In my mind it's simpler and less confusing to the reader. This approach could be applied to the relevant Table footnotes as well.
A: Thank you for pointing out the problem. According to your suggestion, the over complicates of "without an oral calcium bolus containing 90g of calcium.... " has been simplified to "without an oral calcium bolus" in the revised manuscript.
C: L112 reword: Rations were subsequently fed to individual cows.
A: Thank you for pointing out the problem. According to your suggestion, the sentence “Rations was then fed to individual cows subsequently.” has been revised to “Rations were subsequently fed to individual cows.” in the revised manuscript.
C: L242 typo: without.
A: Thank you for pointing out the problem. According to your suggestion, the word “wothout” has been revised to “without” in the revised manuscript.
C: L262 - delete "and"; perhaps some indication of normal blood calcium concentration in the sentence to provide context with respect to subclinical hypocalcemia threshold.
A: Thank you for pointing out the problem. According to your suggestion, the word “and” has been deleted in the revised manuscript. In addition, the sentence “Subclinical hypocalcemia changed with the threshold selected, and subclinical hypocalcemia might happened in control cows when the threshold was iCa≤1.0 mM and tCa≤2.0 mM. Nevertheless, when the threshold was tCa<2.15 mM, both control and 25D cows had potential to be diagnosed with subclinical hypocalcemia.” has been added in the revised manuscript.
C: L296 suggest replacing delivery with calving.
A: Thank you for pointing out the problem. According to your suggestion, the word “delivery” has been revised to “calving” in the revised manuscript.
C: Tables: the treatment headings could be clearer for readers I think. My suggestion would be: Control, Ca+VitD, 25D and Ca+25D.
A: Thank you for pointing out the problem. According to your suggestion, the treatment headings have been revised to “Control, Ca+VitD, 25D and Ca+25D” in the revised manuscript.
We tried our best to improve the English writing and made some changes in the revised manuscript. These changes will not influence the content and framework of the manuscript. We appreciate for editor’s and reviewers’ critical comments and thoughtful suggestions for our manuscript, and hope that the revised manuscript will meet the standard of Animals.
Once again, thank you very much for your comments and suggestions.
Sincerely Yours,
Yonggen Zhang

Reviewer 2 Report
First of all, I congratulate the authors of this article and for the innovations brought by the different laboratory analyzes. However, I made some suggestions for changes that were noted in the text of the article. I also suggest that the authors redo the citations, placing the year of publication in the text is not allowed, only the number of the article.
I suggest improving the discussion. There is little discussion of the data obtained. Improve tables by making them self-explanatory. Eg VD?
It was possible to observe that the authors had already worked with the addition of 25-hydroxyvitamin D3, and even associated it with a DCAD diet, which stimulated the consumption of animals, thus concluding that the work was carried out on a subject that already has similar data . What do the authors have to say about this?

Author Response
Reviewer: 2
C: First of all, I congratulate the authors of this article and for the innovations brought by the different laboratory analyzes. However, I made some suggestions for changes that were noted in the text of the article. I also suggest that the authors redo the citations, placing the year of publication in the text is not allowed, only the number of the article.
A: Thank you for pointing out the problem. According to your suggestion, the citations has been revised according to author guidelines in the revised manuscript.
C: I suggest improving the discussion. There is little discussion of the data obtained. Improve tables by making them self-explanatory. Eg VD?
A: Thank you for pointing out the problem. According to your suggestion, the sentence “Subclinical hypocalcemia changed with the threshold selected, and subclinical hy-pocalcemia might happened in control cows when the threshold was iCa≤1.0 mM and tCa≤2.0 mM. Nevertheless, when the threshold was tCa<2.15 mM, both control and 25D cows had potential to be diagnosed with subclinical hypocalcemia within 24 hours postpartum.”; “postpartum feeding 25-hydroxyvitamin D3 was important to maintain serum concen-trations of 25-hydroxyvitamin D3 and tCa during postpartum period due to higher efficiency conversion to 1,25-dihydroxyvitamin D3.”; “In the present study, the treatments did not influence plasma concentrations of energy metabolites, implying that higher milk production caused by treatments did not lead to more serious negative energy balance.”; “In the present study, feeding 25-hydroxyvitamin D3 decreased the concentrations of IL-6 and TNF-α during postpartum period, which was perhaps due to the depression of systemic inflammatory reactions.” has addedd in the revised manuscript. According to your suggestion, the “VD = the effect of the source of vitamin D3, Ca = the effect of oral Ca bolus.” has added to footnote in all tables in the revised manuscript.
C: It was possible to observe that the authors had already worked with the addition of 25-hydroxyvitamin D3, and even associated it with a DCAD diet, which stimulated the consumption of animals, thus concluding that the work was carried out on a subject that already has similar data. What do the authors have to say about this?
A: Thank you for pointing out the problem. Low-calcium or acidogenic diets, dietary vitamin D nutrition management, injectable vitamin D analogs, and postpartum oral bolus Ca or injectable calcium supplements had a potential to decrease the incidences of clinical or subclinical hypocalcemia to a large extent, nevertheless, subclinical hypocalcemia has remained a prevalent issue in the cattle industry and disturbs the majority of transition dairy cows. Although each of the strategies has demonstrated effectiveness for prevention of clinical milk fever, it also is important to keep in mind that further research is needed in these areas, along with exploration of potential combined or new avenues to decrease risk of subclinical hypocalcemia. Indeed, the similar measurements with previous projects were found in present study. Nevertheless, the lactation performance and blood minerals level in short- or long-term were important indicators to evaluate the implementation of effective prevention strategies. Moreover, liver indicators, antioxidant capacity, immunoglobulins and proinflammatory cytokines were further analyzed in the present study with respect to beneficial and potential adverse effects of present strategy. Another critical aspect is a thorough understanding of the contribution of lactation and health of postpartum dairy cows. If my answer does not solve your doubts about this experiment, we will continue to explain for you.
C: Strategies that can improve calcium mobilization or increase blood calcium concentration should be able to prevent hypocalcaemia.
A: Thank you for pointing out the problem. According to your suggestion, the “improve” has revised to “prevent” in the revised manuscript.
C: Define these acronyms in a foot note in all tables.
A: Thank you for pointing out the problem. According to your suggestion, the “VD = the effect of the source of vitamin D3, Ca = the effect of oral Ca bolus.” has added to footnote in all tables in the revised manuscript.
C: I suggest that you make a conclusion focused on medium production animals. Because, high production cows (> 30 kg / day), possibly, the results would be different.
A: Thank you for pointing out the problem. According to your suggestion, the “medium-production dairy cows” has added in the revised manuscript.
We tried our best to improve the English writing and made some changes in the revised manuscript. These changes will not influence the content and framework of the manuscript. We appreciate for editor’s and reviewers’ critical comments and thoughtful suggestions for our manuscript, and hope that the revised manuscript will meet the standard of Animals.
Once again, thank you very much for your comments and suggestions.
Sincerely Yours,
Yonggen Zhang
